# Asymmetric cryo-EM reconstruction of phage MS2 reveals genome structure *in situ*

Roman I. Koning[1,2], Josue Gomez-Blanco[3], Inara Akopjana[4], Javier Vargas[3], Andris Kazaks[4], Kaspars Tars[4], José María Carazo[3] & Abraham J. Koster[1,2]

In single-stranded ribonucleic acid (RNA) viruses, virus capsid assembly and genome packaging are intertwined processes. Using cryo-electron microscopy and single particle analysis we determined the asymmetric virion structure of bacteriophage MS2, which includes 178 copies of the coat protein, a single copy of the A-protein and the RNA genome. This reveals that *in situ,* the viral RNA genome can adopt a defined conformation. The RNA forms a branched network of stem-loops that almost all allocate near the capsid inner surface, while predominantly binding to coat protein dimers that are located in one-half of the capsid. This suggests that genomic RNA is highly involved in genome packaging and virion assembly.

[1] Department of Molecular Cell Biology, Leiden University Medical Center, P.O. Box 9600, 2300 RC Leiden, The Netherlands. [2] Netherlands Centre for Electron Nanoscopy, Institute of Biology Leiden, Leiden University, Einsteinweg 55, 2333 CC Leiden, The Netherlands. [3] Biocomputing Unit, Centro Nacional de Biotecnología, Consejo Superior de Investigaciones Científicas (CNB-CSIC), Darwin 3, Campus Universidad Autónoma, Cantoblanco, Madrid 28049, Spain. [4] Biomedical Research and Study Centre, Ratsupites 1, LV-1067 Riga, Latvia. Correspondence and requests for materials should be addressed to R.I.K. (email: r.i.koning@lumc.nl).

**B**acteriophage MS2 (ref. 1) is a species of the *Levivirus* genus in the *Leviviridae* family of small, positive-sense, single-stranded ribonucleic acid (RNA) bacteriophages that infect their host via adsorption to bacterial pili. The MS2 virion consists of an RNA genome, encapsidated by a $T = 3$ shell, containing coat protein (CP) and a single copy of the maturation or so-called A-protein (AP), which attaches to the viral RNA and binds the host receptor. The MS2 genome is one of the smallest known, comprising just 3,569 nucleotides, and was the first genome—of any life form—to be completely sequenced[2]. The genome encodes four proteins: maturation, coat, lysis and replicase, which are translated at different levels and time points during the bacteriophage life cycle. The RNA adopts a specific secondary structure with many double-stranded regions, which play essential roles in translational regulation and replication[3]. Extensive research has shown that access to ribosomal binding sites for translation is regulated by short and long range base-pairing[4] and binding of RNA stem-loops (SLs) to CP dimers. More specifically, maturation gene translation can only take place during the first stages of synthesis of a new plus strand[5]. The start of the lysis gene is suppressed by a local hairpin and accessed occasionally by ribosomal back scanning and frame shifting of ribosomes that finished translation of the *CP* gene[6]. Expression of the replicase gene, and therefore replication, is controlled by base pairing with a coding region of the CP[7,8], and downregulated by binding of a SL containing the start codon, called the 'translational operator' (TR), to a CP dimer $(CP_2)$[9]. In addition, the RNA secondary structure is involved in replicase and AP binding and resistance to host RNAses[3].

Although MS2 capsids spontaneously assemble *in vitro* from CP alone at high enough concentration[10], its RNA plays an important role in virion formation *in vivo*. Encapsidation of its own genome depends on presence of the AP[11], which specifically binds two RNA sequences at the 3′–untranslated region (nucleotides 388–414) and the 5′-maturation region (nucleotides 3,510–3,527)[12]. Second, capsid formation is highly promoted by interaction of the TR binding to $CP_2$ (ref. 9). Third, specific SL–$CP_2$ interactions form packaging signals (PSs) promote genome packaging[13,14].

Atomic models of the MS2 capsid[15], capsid protein dimer[16] and the translation suppressing 19-nucleotide RNA TR hairpin loop bound to a $CP_2$ (ref. 17) were determined using X-ray crystallography, revealing sequence-specific RNA–protein interactions essential for binding. The structure of aptamers showed that multiple RNA sequences can bind to MS2 $CP_2$ (refs. 17–22). Icosahedral cryo-electron microscopy (cryo-EM) reconstructions of the MS2 virion, including genome and AP, showed that the operator–$CP_2$ interaction is not unique, and that the whole genome is highly connected to the capsid via SLs[23,24]. Asymmetric single particle[24] and tomography[25] cryo-EM reconstructions of the MS2 virion suggested that the genome within MS2 might adopt a specific tertiary structural conformation, although the resolution of these maps was too low ($\sim$4–6 nm) to resolve any recognizable tertiary RNA structures.

Following these investigations, we here determined using high-resolution single particle cryo-EM the asymmetric structure of bacteriophage MS2, to a resolution of 8.7 Å. The map outlines the AP, which replaces one CP dimer. Moreover, it shows an ordered genome that is shaped as a branched network of connected RNA stem-loops, of which the majority interacts with the inside of the capsid. The RNA–CP interactions are primarily located on one side of the capsid, which might have consequences for genome packaging and virion assembly.

## Results

**Asymmetric structure**. The outside of our asymmetric EM map of the MS2 virion (Fig. 1a) at 8.7 Å (EMD-3403) is similar to the previously reported capsid protein X-ray structure (protein data bank (pdb) entry: 2MS2) (ref. 15 and our icosahedrally averaged cryo-EM capsid structure at 4.1 Å (EMD-3402) (Supplementary Figs 1 and 2, Supplementary Movie 5), showing the characteristic openings in the capsid at the five- and three-fold symmetry axes and the small protrusions, formed by amino acid loops, connecting beta-sheet strands A and B, on the capsid surface (dark blue in Fig. 1a,b). More importantly, in the asymmetric virion structure the single copy of the AP is clearly resolved in the capsid (yellow, Fig. 1a–d, Supplementary Movie 1) and replaces one CC conformer-type $CP_2$ in the capsid, which thus contains 178 copies of the CP. The AP forms a 9 nm long 'handle' ($\sim$1.8 nm in diameter) extending outwards with a $\sim$30° angle from the surface from a two-fold symmetry axis position in the capsid, extending over a neighbouring hole at a three-fold axis.

**The RNA**. The inside of our asymmetric MS2 map reveals that the genome within the MS2 capsid has a defined tertiary structure, forming an intricate three-dimensional (3D) branched network of interconnected SLs (Fig. 1c–e, Supplementary Movie 2). The RNA density accounts for 95% of the calculated mass of the full genome. A less ordered region in the map can explain the missing 5%. In total, 59 SL structures were discriminated in the genome, which were all connected to each other, of which the majority (53, 90%) ended near the capsid (Supplementary Fig. 3), while 6 ends were centrally located. To quantify and assess the PSs, the interactions of the RNA SL with $CP_2$, TR-$CP_2$ X-ray model (PDB entry: 1ZDH) was fitted into the EM density map. In total, 44 SLs (83% of the SLs near the capsid and 75% of all identified SLs) ended at a $CP_2$ RNA binding site, 2 SLs interacted with the AP (Fig. 1d) and 7 were not located near a $CP_2$ interface. Close examination of these 44 binding sites showed that at 23 sites the RNA from the X-ray model fit the EM RNA density (Supplementary Fig. 4), with some of them showing in detail the interactions between nucleotides A $-4$ and A $-10$ in the 19-nucleotide RNA chain of the RNA hairpin loop (1ZDH) that both make contact to Thr45 and Ser47 (not shown) located in the beta-sheets of different CP molecules (Fig. 1f). The observed amounts of SLs (59) and PSs (between 23 and 44) match earlier estimates, predicting 53 SLs and 35 PSs, that were based on the predicted RNA secondary structure, obtained by phylogenetic analysis and experimental probing of RNA in solution, and analysis of potential binding SLs[14]. Our results show that the majority of the many SL RNA structures that are present in the genome of MS2 bind the $CP_2$ in the capsid. Since these SLs all have different predicted sequences, as a result this suggests that a wide variety of different RNA sequences actually bind to $CP_2$ *in situ* in an overall similar conjunction as the TR, similar to what was observed with several aptamer–$CP_2$ interactions[18–20]. At several sites RNA hairpin-EM densities were also observed near the capsid but in deviating conformations or sites, often rotated (7 cases) or shifted (7 cases), compared with the TR structure (data not shown). In addition, several stem structures were sideways associated to $CP_2$ sites. Therefore alternative, non-sequence specific, binding modes of the RNA to the $CP_2$ likely exist, including binding to other amino acids in the capsid.

The obtained resolution of the EM map is sufficient to visualize the double-stranded (ds) RNA SLs, including its helical nature; however, single RNA strands are not clearly resolved and therefore it was not possible to trace predicted secondary structures into the 3D map and to allocate predicted SLs in the MS2 genome into the density[14,26]. To investigate all individual SL–$CP_2$ interactions in the genome to the observed SLs in the EM map a higher resolution structure would be

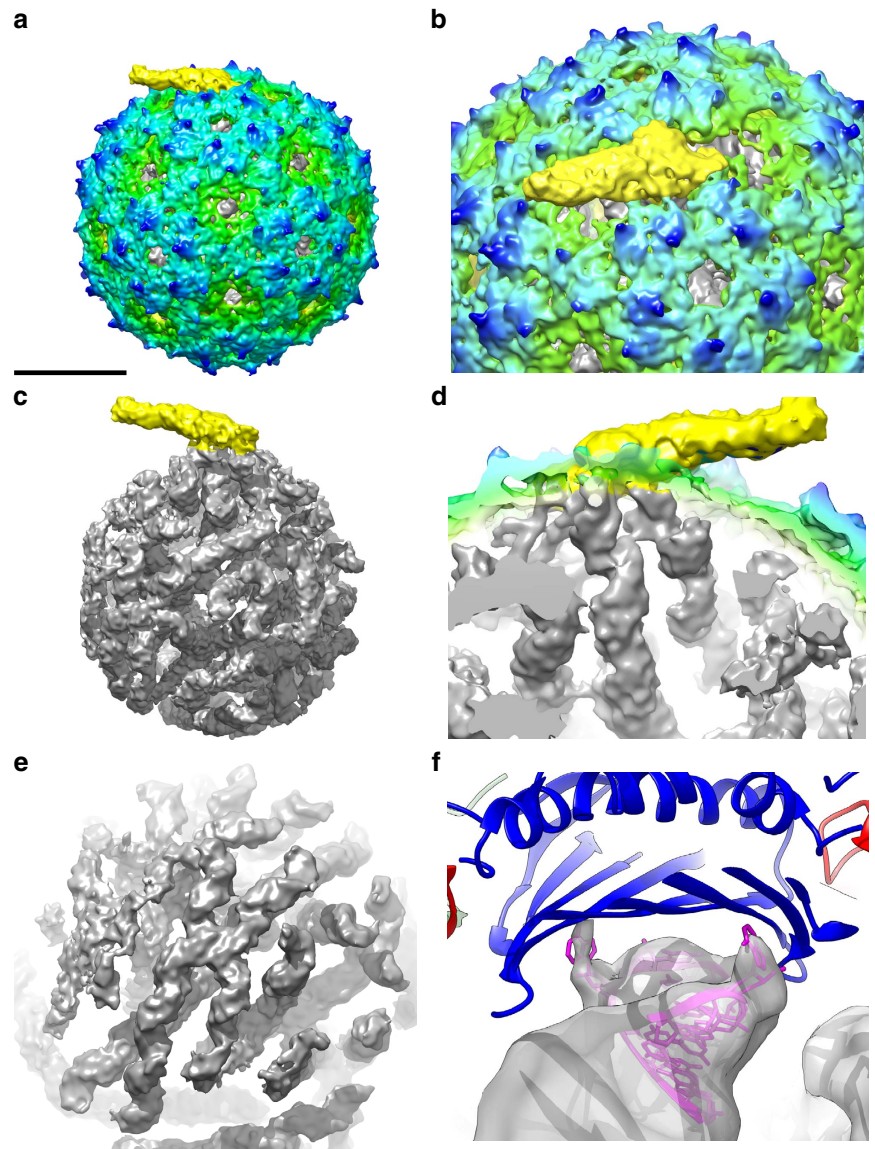

**Figure 1 | The asymmetric reconstruction of bacteriophage MS2.** Asymmetric structure of bacteriophage MS2 (green–blue radially coloured) shows the AP (yellow) (**a**), which replaces one CP dimer (**b**). Inside the protein capsid a structured genome (grey) is present (**c**) that is connected to the AP (**d**). The reconstruction shows the double-stranded helices in the stem loop structures (**e**). At some positions individual NA's connecting to the capsid are resolved, as shown by fitting of the X-ray structure of the 19-nucleotide TR (magenta) bound to the capsid (blue) (pdb:1ZDH) in the EM density (grey) (**f**). Scale bar is 100 Å.

required. Nevertheless, the two regions known to connect to the AP[12] could putatively be allocated in the EM map. The 3′ RNA end, including nucleotides 3,510–3,527, forms a very specific repeat of short SLs that binds adjacently along the AP, while near the 5′ RNA the SL formed by nucleotides 388–414 binds the AP (Fig. 2a, Supplementary Movie 3).

**RNA–protein interactions**. While the AP ensures specific packaging of its coding RNA and TR–$CP_2$ interactions promote capsid formation, the question is how the SL–$CP_2$ interactions that we observed here influence virion assembly. To explore a potential role of the RNA genome structure in this process we investigated the distribution of PSs, interactions of SLs with $CP_2$, over the capsid. From the 89 $CP_2$, 44 (49%) have a connecting SL, 33 (37%) has crossing (ds) RNA density, while 9 (10%) does not have any adjacent density. Notably, these 44 PSs are distributed unevenly over the capsid, being localized predominantly on one

side of the capsid (Supplementary Fig. 3). This uneven distribution was even more pronounced for the 23 RNA SLs of which the density matched the TR X-ray model (Supplementary Fig. 4). Of these, 19 (82%) were bound to dimers that were located on one of half of the capsid in only three $CP_2$ pentamers (Fig. 2b, Supplementary Movie 4). These pentamers were adjacent to the AP and its two binding SLs.

This uneven distribution of SL–$CP_2$ interactions supports a two-step encapsidation model[13,27] in which RNA condensation proceeds full capsid formation. The role of the asymmetric distribution of RNA–protein interactions could be two-fold. Multiple SLs could increase efficient capsid formation by $CP_2$ recruitment from the surroundings to form the first $CP_2$ pentamers, adding both efficiency and localization to $CP_2$–$CP_2$ interactions that drive coat formation. Alternatively, $CP_2$ could induce SLs formation and condensation of the MS2 RNA, thereby inducing genome compaction during encapsidation. These two

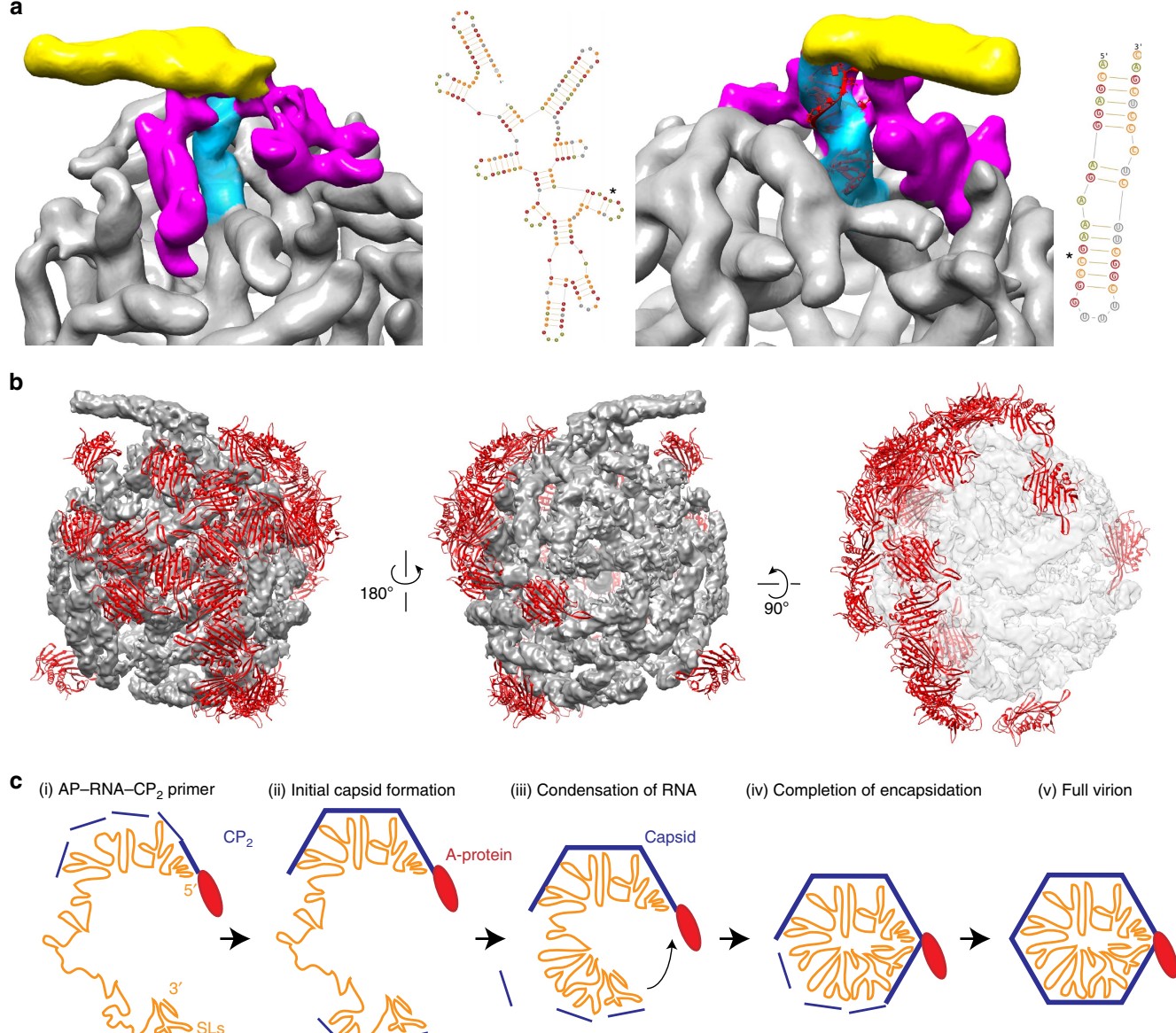

**Figure 2 | Proposed RNA densities of known AP binding sites.** Left: nucleotides 3,510–3,527 (magenta) and the RNA secondary structure prediction (asterisk marks binding site). Right: nucleotides 388–414 (blue) with fit in density (red) and the secondary structure of the SL (**a**). Asymmetric distribution of SL–$CP_2$ interactions; the 23 capsid dimers (red) to which the bound RNA (grey) SLs fit the TR are predominantly localized at one side of the capsid (**b**). Proposed model for MS2 virion assembly: AP–RNA–$(CP_2)_n$ complex forms primer (i) to which $CP_2$ dimers bind and start to form the capsid (ii), $CP_2$ is both being recruited by existing SLs and induce new SLs in the RNA (iii) $CP_2$ induced refolding and binding of the 5′ end AP binding site results in condensation of the RNA (iv) that enables efficient packaging and formation of a stable virion (v) (**c**).

possible functions are not mutually exclusive and also support the suggested formation of new SLs in the MS2 genome[14].

## Discussion

We reconstructed the MS2 virion, revealing the AP and the single conformation of the RNA genome *in situ*. The genome is intimately and asymmetrically linked to the capsid. The presence of a single AP, breaking the symmetry in the capsid, in MS2 is exceptional among viruses and might play an important role in the uniquely structured genome, which might not appear in other (small) viruses. Even so structures of several other viruses have shown hints of (partly) ordered (ds)RNA[28,29]. It remains to be seen whether asymmetric cryo-EM single particle reconstructions of other viruses would reveal similar genome ordering inside

virions. The potential of cryo-EM to explore asymmetric structure determination of complete virions, including their genome, would provide unprecedented details on viral RNA structures, RNA–protein interactions and insight into viral assembly, which is valuable knowledge for drug design by targeting disruptions of viral genome folding and packaging.

## Methods

**Purification of phages.** An overnight culture of *E. coli* strain XL1 blue was grown at 37 °C in LB medium until an $OD_{600}$ of 0.5. Calcium chloride was added to a final concentration of 2 mM and the cells were infected with phage MS2 at a multiplicity of 10 and incubated for another three hours. Lysates with phage titres of approximately $1 \times 10^{-12}$ were used for purification. Cellular debris was removed by centrifugation and phage concentrated by ultrafiltration using 15 ml 100 kDa cutoff Amicon concentrators. Phage was further purified by gel-filtration

on sephacryl S500 column. Fractions were inspected for purity on coomassie-stained SDS-PAGE gel and negative staining EM.

**Specimen preparation.** Aliquots of purified MS2 were applied to glow-discharged holey carbon film supported by cupper grids (Quantifoil R2/2) after glow discharging with negative polarity for 1 min at 30 mA using a K950X carbon coater (Emitech). Grids were vitrified by plunging into a liquid propane/ethane mixture (2:1 v/v), which was cooled by liquid nitrogen. Samples were plunged using a Leica EM GP from room temperature and blotted for 1–2 s using filter paper. After vitrification, the grids were stored in liquid nitrogen until use.

**Data collection.** Data acquisition was performed on a Titan Krios transmission electron microscope (FEI) operated with Cs correction at 300 keV using EPU automated single particles acquisition software (FEI). Seven frames per images were recorded on a back-thinned Falcon II detector at a nominal magnification of 59,000 × with a sampling size of 1.14 Å per pixel.

**Image processing.** Image processing was performed using Scipion platform (http://scipion.cnb.csic.es), which is an integrative image processing framework that currently mainly uses Xmipp (http://xmipp.cnb.csic.es/)[1], Relion (http://www2.mrc-lmb.cam.ac.uk/relion/index.php/Main_Page)[2], Spider (http://spider.wadsworth.org)[3] and EMAN (http://blake.bcm.edu/emanwiki/EMAN2)[4] packages. Graphics were produced by UCSF Chimera (http://www.cgl.ucsf.edu/chimera)[5]. All movies were aligned using Optical Flow approach[6], while contrast transfer functions (CTFs) were estimated using CTFFIND3 (ref. 7), and were used to select the best quality micrographs. A total of 22,441 particles were picked automatically using Xmipp[8], which were further screened[9] extracted and normalized. First, an icosahedrally symmetrized reconstruction was calculated. Particles were classified using 2D reference-free Relion approach, and a subset of 18,977 particles was selected from the best classes. Then, these particles were used for Relion 3D refinement, using standard parameters for viral particles, as icosahedral symmetry and gold-standard approach. A second round of refinement was performed, now without applying any symmetry, using Xmipp projection matching. As initial model the final icosahedrally symmetrized map obtained with the previous refinement of Relion was used, filtered to 25 Å. The first four iterations were performed as global refinements decreasing the sampling search angle. Also, the whole particle set was randomly split in two halves. Each subset was refined with the same conditions as the whole set.

**Data analysis and visualization.** Fitting and visualization was performed using UCSF Chimera[5]. Structures of the MS2 capsids, full and missing one dimer, were both created from the X-ray structures of the MS2 capsid protein (pdb: 2MS2 (ref. 10)), and of the MS capsid protein including the operator hairpin loop with (pdb model 1ZDH[11]). Capsids were manually aligned with the EM density and fine aligned using the fitmap command, normalized, density maps at the map resolution were created using the molmap command, which were then subtracted from the EM density using the vop command to create a map of the RNA and A-protein only (EMD-3404).

**Data availability.** Density maps of the icosahedral reconstruction, the asymmetric reconstruction, and the RNA + AP map, a difference map from the asymmetric reconstruction and the dimer depleted capsid structure, are available from the Electron Microscopy Data Bank with accession codes: EMD-3402, EMD-3403, and EMD-3404, respectively. The authors declare that all data supporting the findings of this study are available from the corresponding author upon request.

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

## Acknowledgements

We thank Sjoerd van den Worm, Rene Olsthoorn, Janis Rumnieks and Christoph Diebolder for discussions and reading of the manuscript. This research has been executed with support of NeCEN, the Netherlands Centre for Electron Nanoscopy, Leiden, NL and NWO, the Netherlands Organisation for Scientific Research and is partly financed by the European Regional Development Fund of the European Commission. We acknowledge the Spanish Ministry of Economy and Competitiveness through Grants AIC-A-2011-0638, BIO2013-44647- R. This work was funded by Instruct (Grant reference PID: 1114), part of the European Strategy Forum on Research Infrastructures (ESFRI) in collaboration with the Instruct Image Processing Center I2PC. This work was supported by NanoNextNL of the Government of the Netherlands and 130 partners.

## Author contributions

R.I.K. conceived the project, collected and analysed the data and wrote the manuscript, J.G.B and J.V. performed image processing, analysed the data, and wrote the manuscript I.A. and A.K purified the MS2, K.T, J.M.C. and A.J.K. supervised the studies.

## Additional information

**Competing financial interests:** The authors declare no competing financial interests.

