## [Peer review file · Nature Communications]

Reviewers' comments:

Reviewer #1 (Remarks to the Author):

MS2 bacteriophage is an important model system for nucleic acid biology and has given rise to critical RNA reagents. This structure represents an advance over the previous MS2 virion structures in the understanding of the RNA structure and the symmetry-breaking AP insertion into the otherwise icosahedral capsid protein shell. The map itself is likely to be of high interest to many groups, probably even more so than its discussion and interpretation here. Here are my overall comments of the manuscript:

I have no doubt on the correctness of the icosahedral symmetry imposed reconstruction. However, I am less confident of their reconstruction without symmetry. How does one verify the correctness of the map shown in Figure 1 though it has a promising looking FSC plot. Though the qualitative description of the structure without symmetry sounds fine, I don't know if the map is correct simply because the reconstruction procedure is not adequately described? Since the virus assembly is made up of 60 asymmetric units of the capsid proteins, they would dominate the signals in the particle orientation determination with icosahedral symmetry. If their structure is correct, their image processing software must be able to pool out the particle orientation correctly from the non-icosahedral components. Will the same approach be applicable to other phage particles like HK 97 phage to determine the DNA structure? The authors should discuss in such context because their DNA arrangement is different from other phage particles. A more detailed description of the image processing protocol will be helpful so that such an experiment may be replicated or tried by others on other virus particles. The figure 1 does not allow me to judge how the capsid protein structure matches with the crystal structure at the reported resolution. There is no scale bar in figure 1. What is the width of the RNA and the distance of the apparent major and minor groove of the RNA respectively? Can the volume density interpreted to be RNA account for all the RNA in the virion? The methods section states, "The particles were re-processed using Xmipp projection matching, using the final map of the previous refinement as initial model and without any symmetry, i.e. with C_1 symmetry." The authors should specify whether such re-processing maintained independence of half-datasets or whether instead some final Relion map was used as an initial model, and in the latter case how overfitting was avoided.

An image and the image quality assessment will be useful for the readers to evaluate the data quality.

What are the details of RNA-protein interactions referred in Fig. 1E. Please elaborate.

The asymmetric and icosahedral maps should be deposited in the field-standard Electron Microscopy Databank (EMDB) as mandated in the cryoEM field, and accession numbers should be reported in the manuscript.

The introduction mentions most structural work but omits reference to Toropova et al 2008, which seems to be the most critical recent data and model to refer to.

Perhaps it would be useful to the more general readership of Nature Comm to note briefly how atypical MS2 is, and how these mechanisms are not generalizable to other virus families.

Were any insights derived concerning the organization of the RNA towards the center of the capsid, away from SL-CP2 interactions?

It is a misnomer to entitle one subsection in the "Results" section "Encapsidation process," as no direct results on encapsidation are presented. This subsection states in part, "The numerous SL-CP2 interactions obviously stabilize the virion. Their uneven distribution suggests that they also have a function during capsid formation, which could be two-fold. Multiple SLs could increase

efficient capsid formation by CP2 recruitment from the surroundings to form the first CP2 pentamers, adding both efficiency and localization to CP2-CP2 interactions that drive coat formation. Alternatively CP2 could induce SLs formation and condensation of the MS2 RNA, thereby inducing genome compaction during encapsidation." That the numerous SL-CP2 interactions stabilize the virion should be backed with references or inference, as it is not obvious. It is unclear why their uneven distribution should suggest a function during capsid formation.

Beckett & Uhlenbeck have shown that isolated TR moieties can catalyze T=3 condensation, and this was later recapitulated by Stockley and others. Therefore it seems to me more likely that the uneven distribution of SL-CP2 interactions is caused by the intrinsic folding of the MS2 genome but does not have any specific effect on capsid morphology. Also, though the role of SL-CP2 interactions is established in targeting the genome to the capsid interior, is it clear that further compaction of the genome is required during encapsidation, given its already condensed nature?

S3 legend "left" should be "left"

Supplementary movie M2, the word "the" is redundant in the second line.

Reviewer #2 (Remarks to the Author):

König et al report an asymmetric structure of the RNA bacteriophage MS2 at 10.5Å resolution (as well as a 5Å symmetrized structure) in which much of the detail of the A protein and RNA are clearly visible. The paper is of fundamental importance in that it shows that the RNA is organized in closely similar ways from particle to particle in MS2, an outstanding question for simple RNA viruses. The structure also largely validates the concept of repeated RNA stem-loop binding sites (based on RNA secondary structure prediction) for the capsid protein put forward by Stockley and Twarick. The authors indicate that they can clearly identify RNA secondary structures that are largely stem-loops, like the RNA operator sequence previously determined and demonstrated at high resolution by crystallography. The resolution is not sufficient, however, to determine the connections between the stem-loops, thus on the basis of this structure we do not know what RNA sequences are participating in the different modes of interaction with the capsid protein. The fact that there is a dominance of stem-loop interactions that mimic that of the operator sequence interaction with the capsid protein is extremely interesting and the fact that this particular type of interaction is asymmetrically distributed in the capsid has major implications for assembly as discussed by the authors.

Overall the paper is well written and illustrated with figures and movies and the technical aspects appear well done as far as can be told from the rather brief methods section. It would be helpful if the authors would address the following points.

1. What role did the A protein play in successfully determining the asymmetric structure? Would it have been possible to determine the asymmetric structure based only on the asymmetry of the RNA density? This of course is critical for RNA viruses that have truly symmetric capsids. Would the signal from the RNA alone in these viruses allow asymmetric reconstructions? The special nature of MS2 (A protein) may lead to the repeated RNA structure from particle to particle that may not be the case generally. It would be helpful to have a couple of sentences in the conclusions regarding this point.
2. The authors do not mention the concept of the Hamiltonian RNA path developed by Twarick and Stockley. Since the A protein has a sequence specific interaction with the RNA, allowing at least one point in the structure where the RNA sequence is known, might some of the previously developed theory be of use in interpreting their structure?
3. The asymmetry of the stem-loop binding region in the capsid supports the hypothesis that the asymmetric structure, indeed, reveals a true repeated organization of RNA, but is it possible that

different stem-loop RNA sequences bind to the same capsid protein site, therefore contributing to the difficulty in determining the connectivity?

Reviewer #1 (Remarks to the Author): MS2 bacteriophage is an important model system for nucleic acid biology and has given rise to critical RNA reagents. This structure represents an advance over the previous MS2 virion structures in the understanding of the RNA structure and the symmetry-breaking AP insertion into the otherwise icosahedral capsid protein shell. The map itself is likely to be of high interest to many groups, probably even more so than its discussion and interpretation here.

Here are my overall comments of the manuscript:

I have no doubt on the correctness of the icosahedral symmetry imposed reconstruction. However, I am less confident of their reconstruction without symmetry:

- How does one verify the correctness of the map shown in Figure 1 though it has a promising looking FSC plot. Though the qualitative description of the structure without symmetry sounds fine, I don't know if the map is correct simply because the reconstruction procedure is not adequately described?

With EM maps of completely new structures that are only resolved below 7 to 8 Å resolution -in which protein structures like alpha helices and beta sheets are not resolved- it is difficult to check the correctness of the map and therefore there is always a chance that the reconstruction is not correct. A good-looking FSC plot is no guarantee for map correctness.

We very recently have obtained a work-in-progress reconstruction with improved resolution to 7.3 Å that confirms the correctness of the structure by observation of the alpha helices on the phage surface while the RNA structure is essentially the same as in the 10.5 Å map, since not many extra features become visible in the RNA at that resolution increase.

Rebuttal Figure 1. Surface rendering of MS2 at 7.3 Å resolution shows the alpha helices on the phage surface.

We have restructured the Materials and Methods section with a more detailed description on the reconstruction procedure, which should make it possible to better follow the reconstruction. Mark that no special image processing procedures were followed and that the asymmetric map was refined from an initial low-pass filtered icosahedral reconstruction.

Note that the reconstruction is performed with Scipion, which allows easy sharing of complete image processing workflow and data.

- Since the virus assembly is made up of 60 asymmetric units of the capsid proteins, they would dominate the signals in the particle orientation determination with icosahedral symmetry. If their structure is correct, their image processing software must be able to pool out the particle orientation correctly from the non-icosahedral components.

This is correct. The projection images of MS2 particle are a combination of the signals from the icosahedral capsid protein and the non-symmetric RNA. The RNA mass is $\sim 33\%$ of the total weight of the virion and therefore gives high enough signal to guide the 2D classification and 3D alignment during non-symmetrical image processing of the whole virion. Even when starting from an initially icosahedral reconstruction.

To test reviewers' remark, we subtracted the icosahedral capsid from the 2D images and reconstructed the RNA density alone, using four separate classes. This shows that reconstructing the asymmetric density alone indeed also leads to the same overall RNA structure. The four classes all are similar to the reconstruction that was made with the capsid present, albeit at lower resolution (since there are less particles per class). Also, all classes are very similar, apart from some variation (asterisk in figure) in the less well-defined stem-loop in the original map.

Note that the density of the A-protein now is not in contact with the RNA (missing density arrow in figure) since a full capsid with 180 coat protein dimers is subtracted, and that the CC dimer that is replaced by the A-protein was not removed before subtraction.

Rebuttal Figure 2. Surface rendering of MS2 RNA. Left (green) a difference map of the asymmetric MS2 virion reconstruction with subtracted capsid (178 coat protein dimers, with missing CC dimer compared to full capsid) and (right) four different classes of RNA reconstruction from 2D images in which a full MS2 capsid was subtracted (grey, pink, purple, magenta). Black arrows denote missing density of A-protein (because full capsid with 180 dimers were subtracted), white stars denote region of small differences.

- Will the same approach be applicable to other phage particle like HK 97 phage to determine the DNA structure? The authors should discuss in such context because their DNA arrangement is different from other phage particles.

The approach, from an image processing point of view, is not special and can be applied to any virus.

Whether the approach in a biological sense will be applicable depends on the particle. Note that MS2 is not a DNA virus but an RNA virus. The secondary RNA structure of MS2 has a distinct non-linear pattern of connected stem-loops, which results in a distinct 3D folding pattern. It is unlikely that DNA viruses have such a patterns, since their genomes are linearly structured, and packaging is likely to be based on non-structure specific layered stacking (as can be seen in EM reconstructions of DNA viruses e.g. T4 phage). Therefore we do not expect that the asymmetrical approach would generally work for DNA viruses. But it is not impossible, and fully depends on the DNA packaging would be similar in all particles.

- A more detail description of the image processing protocol will be helpful so that such experiment may be replicated or tried by others on other virus particles.

We have added a few more details to clarify the materials and methods. However it must be noted that standard image processing methods were used and that the success will mainly depend on the quality of the EM images and the biological structure of the virion particles themselves (since they all should be identical for the reconstruction to converge to a single structure).

- The figure 1 does not allow me to judge how the capsid protein structure matches with the crystal structure at the reported resolution. There is no scale bar in figure 1.

We extended supplemental figure 1 showing a fitted structure of the capsid (pdb entry 2MS2) also with the asymmetric reconstruction. A scale bar is added to figure 1A.

- What is the width of the RNA and the distance of the apparent major and minor groove of the RNA respectively?

In our 1 nm resolution map the requested measurements are relatively rough, and also depend on the choice of the density

threshold for surface rendering, but we measured a RNA width of ~ 2.2 nm and a major groove distance of ~ 2.0 nm and a minor groove of ~ 1.4 nm.

- Can the volume density interpreted to be RNA account for all the RNA in the virion?

According to our calculations, using the sequences and molecular weights of the RNA and the proteins, and measurements of the protein and RNA volumes in the map, it appeared that the RNA volume density accounts for 95.5% of the genome. The missing 4.5% is expected to originate from averaged out density at the site in the map where one RNA stem loop seems more flexible and the resolution is lower (see also rebuttal figure 2). So we expect all RNA to be present inside the capsid.

It must be said that the calculations highly depend on the average density of protein and RNA, which can vary dependent on protein size (Fischer et al. 2004 prot sci) and type of RNA (single stranded, double stranded), so the calculated value is prone to deviation, and should be seen more as an indication.

We added this value of 95.5% in the manuscript.

- The methods section states, "The particles were re-processed using Xmipp projection matching, using the final map of the previous refinement as initial model and without any symmetry, i.e. with c1 symmetry. " The authors should specify whether such re-processing maintained independence of half-datasets or whether instead some final Relion map was used as an initial model, and in the latter case how overfitting was avoided.

We restructured and extended the Materials and Methods section to clarify this issue. In short, the two refinements were independently split into two halves for the initial icosahedral reconstruction using the asymmetric reconstruction. Over-fitting was avoided by filtering the initial icosahedral map to 2.5 nm before using it as initial model for asymmetric reconstruction.

- An image and the image quality assessment will be useful for the readers to evaluate the data quality.

We added a supplementary figure showing and movie aligned cryo-EM image of vitrified MS2, its FFT and how the CTF was fitted using CTFFIND3.

- What are the details of RNA-protein interactions referred in Fig. 1E. Please elaborate.

The text should read figure 1F, which we changed in the text. The interactions refer to nucleotides A -4 and A -10 in the 19 NT RNA chain of the RNA hairpin loop (1ZDH) that both make contact to Thr45 and Ser47 (not shown) located in the beta-sheets of different coat protein molecules. This information has been added to the text.

- The asymmetric and icosahedral maps should be deposited in the field-standard Electron Microscopy Databank (EMDB) as mandated in the cryoEM field, and accession numbers should be reported in the manuscript.

We deposited the maps of the symmetric (EMD-3402) and asymmetric reconstruction (EMD-3403), as well as the difference map showing only the RNA (EMD-3404) in the EMDB and added the accession numbers in the text.

- The introduction mentions most structural work but omits reference to Toropova et al 2008, which seems to be the most critical recent data and model to refer to.

We have added Toropova et al. 2008 to the manuscript.

- Perhaps it would be useful to the more general readership of Nature Comm. to note briefly how atypical MS2 is, and how these mechanisms are not generalizable to other virus families.

This is a good point. We now extended the part in the conclusions that discusses this case compared to other viruses: "The presence of a single AP copy breaking the symmetry in the capsid of MS2 is exceptional among viruses and might play an important role in the uniquely structured genome, which might not appear in other (small) viruses."

- Were any insights derived concerning the organization of the RNA towards the center of the capsid, away from SL-CP2 interactions?

There were no more insights than the ones that can be extracted from the paper: few SLs that are present in the central part, the

resolution is generally lower in the center (where the RNA is not attached to capsid and in one parts of the center there is flexible RNA part). Importantly to note is that the RNA densities are connected thus forming a continuous structure, also in the center. We added a note on this latter part in the text.

- It is a misnomer to entitle one subsection in the "Results" section "Encapsidation process," as no direct results on encapsidation are presented.

Correct. We have changed this subsection into "RNA-protein interactions".

- This subsection states in part, "*The numerous SL-CP2 interactions obviously stabilize the virion. Their uneven distribution suggests that they also have a function during capsid formation, which could be two-fold. Multiple SLs could increase efficient capsid formation by CP2 recruitment from the surroundings to form the first CP2 pentamers, adding both efficiency and localization to CP2-CP2 interactions that drive coat formation. Alternatively CP2 could induce SLs formation and condensation of the MS2 RNA, thereby inducing genome compaction during encapsidation.*" That the numerous SL-CP2 interactions stabilize the virion should be backed with references or inference, as it is not obvious.

We could find no direct evidence in literature that measured the stability of the MS2 virion and empty capsids, though it was shown that the presence of RNA influences capsid formation. We also rewrote this paragraph and removed the statement.

- It is unclear why their uneven distribution should suggest a function during capsid formation.

The statement as was put in the manuscript indeed needed some clarification and explanation. An uneven distribution as such is no reason for a function in capsid formation, even though the presence of RNA influences formation. Therefore we added the reference of Borodovka (2012) and rewrote this paragraph now starting with referring to earlier data.

The notion that an uneven distribution of SL-CP2 interactions over the capsid might play a role in capsid formation came upon us after reading the articles from Borodavka et al. (2012) and Borodavka et al. (2013) in which a two stage mechanism for RNA compaction and

assembly was presented in which both compaction of RNA and capsid formation were linked. They showed that the formation is a two-step process that involves condensation of the RNA, and particle growth. Together with our results we interpreted this data as put forward in the hypothetical model in Figure 2 D.

- Beckett & Uhlenbeck have shown that isolated TR moieties can catalyze T=3 condensation, and this was later recapitulated by Stockley and others. Therefore it seems to me more likely that the uneven distribution of SL-CP2 interactions is caused by the intrinsic folding of the MS2 genome but does not have any specific effect on capsid morphology.

Indeed, the final asymmetric distribution is fully a consequence of the folding and refolding of the tertiary structure of the genome, and does not have an effect on the final capsid morphology. Isolated TR moieties probably act on the stability and/or structure of capsid protein and thereby catalyze T=3 condensation. However, the full genome has connected TR-like moieties that, during initial capsid formation, can catalyze the nucleation of the first pentamers and hexamers, which needed to create a curved closed capsid surface, by physically bringing protein dimers close together. Therefore the distribution of SLs over the RNA can well affect formation, especially in initial stages. An symmetric distribution will in this case be advantageous for nucleation, initial capsid formation, as in implied in our model.

- Also, though the role of SL-CP2 interactions is established in targeting the genome to the capsid interior, is it clear that further compaction of the genome is required during encapsidation, given its already condensed nature?

Not from our data, but the articles from Borodavka et al. (2012, 2013) show that the RNA indeed compacts further during virion formation.

- S3 legend "keft" should be "left"

We changed the text accordingly.

- Supplementary movie M2, the word "the" is redundant in the second line.

We changed the text accordingly.

Reviewer #2 (Remarks to the Author): Koning et al report an asymmetric structure of the RNA bacteriophage MS2 at 10.5Å resolution (as well as a 5Å symmetrized structure) in which much of the detail of the A protein and RNA are clearly visible. The paper is of fundamental importance in that it shows that the RNA is organized in closely similar ways from particle to particle in MS2, an outstanding question for simple RNA viruses. The structure also largely validates the concept of repeated RNA stem-loop binding sites (based on RNA secondary structure prediction) for the capsid protein put forward by Stockley and Twarock. The authors indicate that they can clearly identify RNA secondary structures that are largely stem-loops, like the RNA operator sequence previously determined and demonstrated at high resolution by crystallography. The resolution is not sufficient, however, to determine the connections between the stem-loops, thus on the basis of this structure we do not know what RNA sequences are participating in the different modes of interaction with the capsid protein. The fact that there is a dominance of stem-loop interactions that mimic that of the operator sequence interaction with the capsid protein is extremely interesting and the fact that this particular type of interaction is asymmetrically distributed in the capsid has major implications for assembly as discussed by the authors.

- Overall the paper is well written and illustrated with figures and movies and the technical aspects appear well done as far as can be told from the rather brief methods section.

We have extended the Materials and Methods section.

It would be helpful if the authors would address the following points:

- What role did the A protein play in successfully determining the asymmetric structure?

We did not specifically test the effect of masking out the A-protein during alignment (which is not trivial). But the role of the A-protein is probably low it represents a small part of the asymmetric structure of MS2. It is the RNA that drives the asymmetric reconstruction, since its mass is much larger than that of the A-protein. (See also rebuttal figure 2 and answer to reviewer #1).

It can be seen that the A-protein does show up in the 2D classes.

Rebuttal Figure 3. 2D class averages from MS2 showing the location of the A-protein (black arrows).

- Would it have been possible to determine the asymmetric structure based only on the asymmetry of the RNA density? This of course is critical for RNA viruses that have truly symmetric capsids.

Yes, see the above answer to earlier questions of reviewer #1 and rebuttal figure 2. We tested this by subtracting the density of the capsid from the images and reconstruct the RNA only. Results show that for MS2 this is technically possible, from an image processing point of view.

Whether that would be possible for other viruses is indeed the question. From a biological point of view this is possible under the (obvious) condition that the RNA adopts a stable conformation inside the virion and does not vary between different particles. It might be so that the presence of the A-protein is necessary to lock the genome in a specific structural conformation. But possibly other viruses have evolved other solutions for this. Additionally, larger viruses will have more degrees of freedom for packaging RNA in their centers and so are less likely to be fully ordered, but incomplete or lower ordering, e.g. near the capsid, might be possible.

- Would the signal from the RNA alone in these viruses allow asymmetric reconstructions?

Yes. See answer to first reviewer (and rebuttal figure 1).

- The special nature of MS2 (A protein) may lead to the repeated RNA structure from particle to particle that may not be the case generally. It would be helpful to have a couple of sentences in the conclusions regarding this point.

Good point. The A protein serves as a unique 'handle' to the RNA which might be necessary for the unique structure to form. And therefore this feature might not be applicable to all viruses.

However, we think there are additional determinants that would predict whether other viruses could have a (partly) unique RNA fold are not depended on having a single anchor point like the A-protein: (i) the viruses should be single stranded RNA viruses that are highly double stranded in nature (ii) they should have many predicted stem-loops or other specific structures that are know to bind to the capsid (iii) the genome and capsid size should be relatively small, since otherwise the freedom in the central part of the virion would not likely be structured. A determinant could be that icosahedral reconstructions should show a similar RNA density pattern to that of MS2, which hints toward having a defined conformation.

We added a sentence on the conclusions of the manuscript stating: "The presence of a single AP copy breaking the symmetry in the capsid of MS2 is exceptional among viruses and might play an important role in the uniquely structured genome, which might not appear in other (small) viruses."

- The authors do not mention the concept of the Hamiltonian RNA path developed by Twarick and Stockley. Since the A protein has a sequence specific interaction with the RNA, allowing at least one point in the structure where the RNA sequence is known, might some of the previously developed theory be of use in interpreting their structure?

The virion map in the current manuscript shows that the Hamiltonian path theory is completely irrelevant for MS2, simply because Hamiltonian path theory implies (i) a linear two-dimensional path (ii) along the capsid dimer surface (iii) visiting all positions only once. In reality, our map shows that (i) the RNA is not linear but highly branched, (ii) does not follow a part over the capsid surface but can cross over, and (iii) there is one site (A-protein) that has multiple binding sites – while also many are not visited. Unequivocal and complete interpretation of the structure will be possible at near-atomic resolution structures, which we are working on.

- The asymmetry of the stem-loop binding region in the capsid supports the hypothesis that the asymmetric structure, indeed, reveals a true repeated organization of RNA, but is it possible that different stem-loop RNA sequences bind to the same capsid protein site, therefore contributing to the difficulty in determining the connectivity?

Yes. Having variations of RNA sequences in the different individual virus particles is not impossible. To some extent it is even likely since viruses are very prone to mutations, and therefore possible mutations in the RNA make that not all particles are the same in every position and thus also some variations will appear in SL-CP2 interactions. It is however likely that this will average out since many different particles are averaged to create a cryo-EM map. Therefore these individual variations are not expected to have an effect on the structure determination. We expect that a near atomic resolution map will enable interpreting the complete structure and the connectivity of the virion directly.

Reviewers' Comments:

Reviewer #1 (Remarks to the Author)

I am pleased with the revised manuscript. The authors have responded well to the reviewers' queries. I have a minor question why 0.5 FSC instead of 0.143 was used for the resolution criterion.

It is an important technology advance in virus structure determination to be able to see organized genome. I recommend for publication as it is.

Reviewer #1 (Remarks to the Author): MS2 bacteriophage is an important model system for nucleic acid biology and has given rise to critical RNA reagents. This structure represents an advance over the previous MS2 virion structures in the understanding of the RNA structure and the symmetry-breaking AP insertion into the otherwise icosahedral capsid protein shell. The map itself is likely to be of high interest to many groups, probably even more so than its discussion and interpretation here.

Here are my overall comments of the manuscript:

I have no doubt on the correctness of the icosahedral symmetry imposed reconstruction. However, I am less confident of their reconstruction without symmetry:

- How does one verify the correctness of the map shown in Figure 1 though it has a promising looking FSC plot. Though the qualitative description of the structure without symmetry sounds fine, I don't know if the map is correct simply because the reconstruction procedure is not adequately described?

With EM maps of completely new structures that are only resolved below 7 to 8 Å resolution -in which protein structures like alpha helices and beta sheets are not resolved- it is difficult to check the correctness of the map and therefore there is always a chance that the reconstruction is not correct. A good-looking FSC plot is no guarantee for map correctness.

We very recently have obtained a work-in-progress reconstruction with improved resolution to 7.3 Å that confirms the correctness of the structure by observation of the alpha helices on the phage surface while the RNA structure is essentially the same as in the 10.5 Å map, since not many extra features become visible in the RNA at that resolution increase.

Rebuttal Figure 1. Surface rendering of MS2 at 7.3 Å resolution shows the alpha helices on the phage surface.

We have restructured the Materials and Methods section with a more detailed description on the reconstruction procedure, which should make it possible to better follow the reconstruction. Mark that no special image processing procedures were followed and that the asymmetric map was refined from an initial low-pass filtered icosahedral reconstruction.

Note that the reconstruction is performed with Scipion, which allows easy sharing of complete image processing workflow and data.

- Since the virus assembly is made up of 60 asymmetric units of the capsid proteins, they would dominate the signals in the particle orientation determination with icosahedral symmetry. If their structure is correct, their image processing software must be able to pool out the particle orientation correctly from the non-icosahedral components.

This is correct. The projection images of MS2 particle are a combination of the signals from the icosahedral capsid protein and the non-symmetric RNA. The RNA mass is $\sim 33\%$ of the total weight of the virion and therefore gives high enough signal to guide the 2D classification and 3D alignment during non-symmetrical image processing of the whole virion. Even when starting from an initially icosahedral reconstruction.

To test reviewers' remark, we subtracted the icosahedral capsid from the 2D images and reconstructed the RNA density alone, using four separate classes. This shows that reconstructing the asymmetric density alone indeed also leads to the same overall RNA structure. The four classes all are similar to the reconstruction that was made with the capsid present, albeit at lower resolution (since there are less particles per class). Also, all classes are very similar, apart from some variation (asterisk in figure) in the less well-defined stem-loop in the original map.

Note that the density of the A-protein now is not in contact with the RNA (missing density arrow in figure) since a full capsid with 180 coat protein dimers is subtracted, and that the CC dimer that is replaced by the A-protein was not removed before subtraction.

Rebuttal Figure 2. Surface rendering of MS2 RNA. Left (green) a difference map of the asymmetric MS2 virion reconstruction with subtracted capsid (178 coat protein dimers, with missing CC dimer compared to full capsid) and (right) four different classes of RNA reconstruction from 2D images in which a full MS2 capsid was subtracted (grey, pink, purple, magenta). Black arrows denote missing density of A-protein (because full capsid with 180 dimers were subtracted), white stars denote region of small differences.

- Will the same approach be applicable to other phage particle like HK 97 phage to determine the DNA structure? The authors should discuss in such context because their DNA arrangement is different from other phage particles.

The approach, from an image processing point of view, is not special and can be applied to any virus.

Whether the approach in a biological sense will be applicable depends on the particle. Note that MS2 is not a DNA virus but an RNA virus. The secondary RNA structure of MS2 has a distinct non-linear pattern of connected stem-loops, which results in a distinct 3D folding pattern. It is unlikely that DNA viruses have such a patterns, since their genomes are linearly structured, and packaging is likely to be based on non-structure specific layered stacking (as can be seen in EM reconstructions of DNA viruses e.g. T4 phage). Therefore we do not expect that the asymmetrical approach would generally work for DNA viruses. But it is not impossible, and fully depends on the DNA packaging would be similar in all particles.

- A more detail description of the image processing protocol will be helpful so that such experiment may be replicated or tried by others on other virus particles.

We have added a few more details to clarify the materials and methods. However it must be noted that standard image processing methods were used and that the success will mainly depend on the quality of the EM images and the biological structure of the virion particles themselves (since they all should be identical for the reconstruction to converge to a single structure).

- The figure 1 does not allow me to judge how the capsid protein structure matches with the crystal structure at the reported resolution. There is no scale bar in figure 1.

We extended supplemental figure 1 showing a fitted structure of the capsid (pdb entry 2MS2) also with the asymmetric reconstruction. A scale bar is added to figure 1A.

- What is the width of the RNA and the distance of the apparent major and minor groove of the RNA respectively?

In our 1 nm resolution map the requested measurements are relatively rough, and also depend on the choice of the density

threshold for surface rendering, but we measured a RNA width of ~ 2.2 nm and a major groove distance of ~ 2.0 nm and a minor groove of ~ 1.4 nm.

- Can the volume density interpreted to be RNA account for all the RNA in the virion?

According to our calculations, using the sequences and molecular weights of the RNA and the proteins, and measurements of the protein and RNA volumes in the map, it appeared that the RNA volume density accounts for 95.5% of the genome. The missing 4.5% is expected to originate from averaged out density at the site in the map where one RNA stem loop seems more flexible and the resolution is lower (see also rebuttal figure 2). So we expect all RNA to be present inside the capsid.

It must be said that the calculations highly depend on the average density of protein and RNA, which can vary dependent on protein size (Fischer et al. 2004 prot sci) and type of RNA (single stranded, double stranded), so the calculated value is prone to deviation, and should be seen more as an indication.

We added this value of 95.5% in the manuscript.

- The methods section states, "The particles were re-processed using Xmipp projection matching, using the final map of the previous refinement as initial model and without any symmetry, i.e. with c1 symmetry. " The authors should specify whether such re-processing maintained independence of half-datasets or whether instead some final Relion map was used as an initial model, and in the latter case how overfitting was avoided.

We restructured and extended the Materials and Methods section to clarify this issue. In short, the two refinements were independently split into two halves for the initial icosahedral reconstruction using the asymmetric reconstruction. Over-fitting was avoided by filtering the initial icosahedral map to 2.5 nm before using it as initial model for asymmetric reconstruction.

- An image and the image quality assessment will be useful for the readers to evaluate the data quality.

We added a supplementary figure showing and movie aligned cryo-EM image of vitrified MS2, its FFT and how the CTF was fitted using CTFFIND3.

- What are the details of RNA-protein interactions referred in Fig. 1E. Please elaborate.

The text should read figure 1F, which we changed in the text. The interactions refer to nucleotides A -4 and A -10 in the 19 NT RNA chain of the RNA hairpin loop (1ZDH) that both make contact to Thr45 and Ser47 (not shown) located in the beta-sheets of different coat protein molecules. This information has been added to the text.

- The asymmetric and icosahedral maps should be deposited in the field-standard Electron Microscopy Databank (EMDB) as mandated in the cryoEM field, and accession numbers should be reported in the manuscript.

We deposited the maps of the symmetric (EMD-3402) and asymmetric reconstruction (EMD-3403), as well as the difference map showing only the RNA (EMD-3404) in the EMDB and added the accession numbers in the text.

- The introduction mentions most structural work but omits reference to Toropova et al 2008, which seems to be the most critical recent data and model to refer to.

We have added Toropova et al. 2008 to the manuscript.

- Perhaps it would be useful to the more general readership of Nature Comm. to note briefly how atypical MS2 is, and how these mechanisms are not generalizable to other virus families.

This is a good point. We now extended the part in the conclusions that discusses this case compared to other viruses: "The presence of a single AP copy breaking the symmetry in the capsid of MS2 is exceptional among viruses and might play an important role in the uniquely structured genome, which might not appear in other (small) viruses."

- Were any insights derived concerning the organization of the RNA towards the center of the capsid, away from SL-CP2 interactions?

There were no more insights than the ones that can be extracted from the paper: few SLs that are present in the central part, the

resolution is generally lower in the center (where the RNA is not attached to capsid and in one parts of the center there is flexible RNA part). Importantly to note is that the RNA densities are connected thus forming a continuous structure, also in the center. We added a note on this latter part in the text.

- It is a misnomer to entitle one subsection in the "Results" section "Encapsidation process," as no direct results on encapsidation are presented.

Correct. We have changed this subsection into "RNA-protein interactions".

- This subsection states in part, "*The numerous SL-CP2 interactions obviously stabilize the virion. Their uneven distribution suggests that they also have a function during capsid formation, which could be two-fold. Multiple SLs could increase efficient capsid formation by CP2 recruitment from the surroundings to form the first CP2 pentamers, adding both efficiency and localization to CP2-CP2 interactions that drive coat formation. Alternatively CP2 could induce SLs formation and condensation of the MS2 RNA, thereby inducing genome compaction during encapsidation.*" That the numerous SL-CP2 interactions stabilize the virion should be backed with references or inference, as it is not obvious.

We could find no direct evidence in literature that measured the stability of the MS2 virion and empty capsids, though it was shown that the presence of RNA influences capsid formation. We also rewrote this paragraph and removed the statement.

- It is unclear why their uneven distribution should suggest a function during capsid formation.

The statement as was put in the manuscript indeed needed some clarification and explanation. An uneven distribution as such is no reason for a function in capsid formation, even though the presence of RNA influences formation. Therefore we added the reference of Borodovka (2012) and rewrote this paragraph now starting with referring to earlier data.

The notion that an uneven distribution of SL-CP2 interactions over the capsid might play a role in capsid formation came upon us after reading the articles from Borodavka et al. (2012) and Borodavka et al. (2013) in which a two stage mechanism for RNA compaction and

assembly was presented in which both compaction of RNA and capsid formation were linked. They showed that the formation is a two-step process that involves condensation of the RNA, and particle growth. Together with our results we interpreted this data as put forward in the hypothetical model in Figure 2 D.

- Beckett & Uhlenbeck have shown that isolated TR moieties can catalyze T=3 condensation, and this was later recapitulated by Stockley and others. Therefore it seems to me more likely that the uneven distribution of SL-CP2 interactions is caused by the intrinsic folding of the MS2 genome but does not have any specific effect on capsid morphology.

Indeed, the final asymmetric distribution is fully a consequence of the folding and refolding of the tertiary structure of the genome, and does not have an effect on the final capsid morphology. Isolated TR moieties probably act on the stability and/or structure of capsid protein and thereby catalyze T=3 condensation. However, the full genome has connected TR-like moieties that, during initial capsid formation, can catalyze the nucleation of the first pentamers and hexamers, which needed to create a curved closed capsid surface, by physically bringing protein dimers close together. Therefore the distribution of SLs over the RNA can well affect formation, especially in initial stages. An asymmetric distribution will in this case be advantageous for nucleation, initial capsid formation, as is implied in our model.

- Also, though the role of SL-CP2 interactions is established in targeting the genome to the capsid interior, is it clear that further compaction of the genome is required during encapsidation, given its already condensed nature?

Not from our data, but the articles from Borodavka et al. (2012, 2013) show that the RNA indeed compacts further during virion formation.

- S3 legend "keft" should be "left"

We changed the text accordingly.

- Supplementary movie M2, the word "the" is redundant in the second line.

We changed the text accordingly.

Reviewer #2 (Remarks to the Author): Koning et al report an asymmetric structure of the RNA bacteriophage MS2 at 10.5Å resolution (as well as a 5Å symmetrized structure) in which much of the detail of the A protein and RNA are clearly visible. The paper is of fundamental importance in that it shows that the RNA is organized in closely similar ways from particle to particle in MS2, an outstanding question for simple RNA viruses. The structure also largely validates the concept of repeated RNA stem-loop binding sites (based on RNA secondary structure prediction) for the capsid protein put forward by Stockley and Twarock. The authors indicate that they can clearly identify RNA secondary structures that are largely stem-loops, like the RNA operator sequence previously determined and demonstrated at high resolution by crystallography. The resolution is not sufficient, however, to determine the connections between the stem-loops, thus on the basis of this structure we do not know what RNA sequences are participating in the different modes of interaction with the capsid protein. The fact that there is a dominance of stem-loop interactions that mimic that of the operator sequence interaction with the capsid protein is extremely interesting and the fact that this particular type of interaction is asymmetrically distributed in the capsid has major implications for assembly as discussed by the authors.

- Overall the paper is well written and illustrated with figures and movies and the technical aspects appear well done as far as can be told from the rather brief methods section.

We have extended the Materials and Methods section.

It would be helpful if the authors would address the following points:

- What role did the A protein play in successfully determining the asymmetric structure?

We did not specifically test the effect of masking out the A-protein during alignment (which is not trivial). But the role of the A-protein is probably low since it represents a small part of the asymmetric structure of MS2. It is the RNA that drives the asymmetric reconstruction, since its mass is much larger than that of the A-protein. (See also rebuttal figure 2 and answer to reviewer #1).

It can be seen that the A-protein does show up in the 2D classes.

Rebuttal Figure 3. 2D class averages from MS2 showing the location of the A-protein (black arrows).

- Would it have been possible to determine the asymmetric structure based only on the asymmetry of the RNA density? This of course is critical for RNA viruses that have truly symmetric capsids.

Yes, see the above answer to earlier questions of reviewer #1 and rebuttal figure 2. We tested this by subtracting the density of the capsid from the images and reconstruct the RNA only. Results show that for MS2 this is technically possible, from an image processing point of view.

Whether that would be possible for other viruses is indeed the question. From a biological point of view this is possible under the (obvious) condition that the RNA adopts a stable conformation inside the virion and does not vary between different particles. It might be so that the presence of the A-protein is necessary to lock the genome in a specific structural conformation. But possibly other viruses have evolved other solutions for this. Additionally, larger viruses will have more degrees of freedom for packaging RNA in their centers and so are less likely to be fully ordered, but incomplete or lower ordering, e.g. near the capsid, might be possible.

- Would the signal from the RNA alone in these viruses allow asymmetric reconstructions?

Yes. See answer to first reviewer (and rebuttal figure 1).

- The special nature of MS2 (A protein) may lead to the repeated RNA structure from particle to particle that may not be the case generally. It would be helpful to have a couple of sentences in the conclusions regarding this point.

Good point. The A protein serves as a unique 'handle' to the RNA which might be necessary for the unique structure to form. And therefore this feature might not be applicable to all viruses.

However, we think there are additional determinants that would predict whether other viruses could have a (partly) unique RNA fold are not depended on having a single anchor point like the A-protein: (i) the viruses should be single stranded RNA viruses that are highly double stranded in nature (ii) they should have many predicted stem-loops or other specific structures that are know to bind to the capsid (iii) the genome and capsid size should be relatively small, since otherwise the freedom in the central part of the virion would not likely be structured. A determinant could be that icosahedral reconstructions should show a similar RNA density pattern to that of MS2, which hints toward having a defined conformation.

We added a sentence on the conclusions of the manuscript stating: "The presence of a single AP copy breaking the symmetry in the capsid of MS2 is exceptional among viruses and might play an important role in the uniquely structured genome, which might not appear in other (small) viruses."

- The authors do not mention the concept of the Hamiltonian RNA path developed by Twarick and Stockley. Since the A protein has a sequence specific interaction with the RNA, allowing at least one point in the structure where the RNA sequence is known, might some of the previously developed theory be of use in interpreting their structure?

The virion map in the current manuscript shows that the Hamiltonian path theory is completely irrelevant for MS2, simply because Hamiltonian path theory implies (i) a linear two-dimensional path (ii) along the capsid dimer surface (iii) visiting all positions only once. In reality, our map shows that (i) the RNA is not linear but highly branched, (ii) does not follow a part over the capsid surface but can cross over, and (iii) there is one site (A-protein) that has multiple binding sites – while also many are not visited. Unequivocal and complete interpretation of the structure will be possible at near-atomic resolution structures, which we are working on.

- The asymmetry of the stem-loop binding region in the capsid supports the hypothesis that the asymmetric structure, indeed, reveals a true repeated organization of RNA, but is it possible that different stem-loop RNA sequences bind to the same capsid protein site, therefore contributing to the difficulty in determining the connectivity?

Yes. Having variations of RNA sequences in the different individual virus particles is not impossible. To some extent it is even likely since viruses are very prone to mutations, and therefore possible mutations in the RNA make that not all particles are the same in every position and thus also some variations will appear in SL-CP2 interactions. It is however likely that this will average out since many different particles are averaged to create a cryo-EM map. Therefore these individual variations are not expected to have an effect on the structure determination. We expect that a near atomic resolution map will enable interpreting the complete structure and the connectivity of the virion directly.